# Comparative transcriptome analysis in two contrasting genotypes for *Sclerotinia sclerotiorum* resistance in sunflower

**Mingzhu Zhao[1], Bing Yi[1], Xiaohong Liu[1], Dexing Wang[1], Dianxiu Song[1], Enyu Sun[1], Liangji Cui[1], Jingang Liu**[1]*, **Liangshan Feng[2]***

**1** Institute of Crop Research, Liaoning Academy of Agricultural Sciences, Shenyang, China, **2** Liaoning Academy of Agricultural Sciences, Shenyang, China

* jinggangliu2022@163.com (JL); fenglsh@163.com (LF)

**Data Availability Statement:** The data sets supporting the results of this article are included within the article and its supporting information files. Raw transcriptome sequence data have been

## Abstract

*Sclerotinia sclerotiorum* as a necrotrophic fungus causes the devastating diseases in many important oilseed crops worldwide. The preferred strategy for controlling *S. sclerotiorum* is to develop resistant varieties, but the molecular mechanisms underlying *S. sclerotiorum* resistance remain poorly defined in sunflower (*Helianthus annuus*). Here, a comparative transcriptomic analysis was performed in leaves of two contrasting sunflower genotypes, disease susceptible (DS) B728 and disease resistant (DR) C6 after *S. sclerotiorum* inoculation. At 24 h post-inoculation, the DR genotype exhibited no visible growth of the hyphae as well as greater activity of superoxide dismutase activity (SOD), peroxidase (POD), catalase (CAT), glutathione-S-transferase (GST), ascorbate peroxidase (APX) and monodehydroascorbate reductase (MDAR) than DS genotype. A total of 10151 and 7439 differentially expressed genes (DEGs) were detected in DS and DR genotypes, respectively. Most of DEGs were enriched in cell wall organisation, protein kinase activity, hormone, transcription factor activities, redox homeostasis, immune response, and secondary metabolism. Differential expression of genes involved in expansins, pectate lyase activities, ethylene biosynthesis and signaling and antioxidant activity after *S. sclerotiorum* infection could potentially be responsible for the differential resistance among two genotypes. In summary, these finding provide additional insights into the potential molecular mechanisms of *S. sclerotiorum*'s defense response and facilitate the breeding of *Sclerotinia*-resistant sunflower varieties.

## Introduction

Sunfower (*Helianthus annuus*), is the third most important oilseed crop in the world, primarily for production of high-quality cooking oil and confectionary products and bird food, among others [1]. The occurrence of *Sclerotinia* and the lack of Sclerotinia-resistant varieties has led to the decline of the planting area and yield in sunflower. *Sclerotinia sclerotiorum* (Lib.) de Bary is a kind of necrotrophic fungus, which infects over 400 species of plants worldwide including important oilseed crops, such as soybean, canola and sunflower [2–4]. Crop

deposited in the NCBI BioProject database and can be accessed under the BioProject ID PRJNA908908 (https://www.ncbi.nlm.nih.gov/bioproject/PRJNA908908).

**Funding:** The study was supported by National Natural Science Foundation of China (32372211), The Presidential Foundation of the Liaoning Academy of Agricultural Sciences (2023MS0505 and 2024YQ0403), China Agriculture Research System of MOF and MARA (CARS-14), and Shenyang Seed Industry Innovation Project (22-318-2-19). The funders had no role in study design, data collection and analysis, decision to publish, or preparation of the manuscript.

**Competing interests:** The authors have declared that no competing interests exist.

productivity and oil quality can be negatively affected by stem rot, root rot and head rot of plant at different developmental stages [5,6]. Fungal pathogens penetrates the host cuticle first forms an infection pad through mechanical pressure or secreting keratinase, then produce oxalic acid, toxic compounds, cell-wall-degrading enzymes and develop water-soaked lesions, tissue necrosis and finally bear sclerotia [7]. Although a few fungicides are available to manage this disease, the low efficiencies, the environmental costs cannot be ignored [8]. Breeding and cultivating resistant varieties are the best strategy to control *S. sclerotiorum* [9,10]. For the effective breeding of *Sclerotinia*-resistant varieties, therefore, it is essential to understand the genetic and molecular basis of the interactions of *S. sclerotiorum* with its host crops.

Pathogen-associated molecular pattern-triggered immunity (PTI) and effector-triggered immunity (ETI) as the major immune systems in plants have been evolved to cope with pathogens [11]. For pathogen recognition, PTI uses cell-surface pattern recognition receptors (PRRs) that respond to pathogen-associated molecular pattern. The early reactions of PTI include calcium influx, production of reactive oxygen species (ROS), and activation of mitogen activated protein kinases (MAPKs) to trigger transcriptional processes regulated by plant transcription factors [12]. PTI plays an important role in plant resistance to *S. sclerotiorum* [13]. However, adapted pathogens able to suppress PTI may be recognized by plant intracellular resistance proteins of the nucleotide-binding and leucine-rich repeat (NLR) family, encoded by resistance (R) genes, leading to ETI [14,15]. ETI triggers a series of defense responses in the host plant, most commonly at the infection site causing programmed cell death and anaphylaxis (thereby limiting the growth, reproduction, and expansion of the pathogen), and this is a very effective way for biotrophs fungi, but for necrotrophs fungi, it can kill the host cells and feed on the contents, while biotrophs complete their life cycle depending on the living host cells [16]. To date, no PRRs or R proteins involved in recognizing *S. sclerotiorum* in host plants have been observed [17,18].

Transcriptional profiling at the site of infection provides an efficient means to dectect the natural epidemiologic cycle among responses between resistant and susceptible varieties [19]. Previous studies characterized the transcriptomic changes in soybean, *Brassica napus*, and *Arabidopsis thaliana* during the defense responses to *S. sclerotiorum*, which involves rapid induction of key pathogen responsive genes, including glucanases, chitinases, peroxidises and WRKY transcription factor family, pathogenesis related proteins, as well as genes related to signal transduction, cellular redox state, cell wall composition and hormone signaling pathways [20,21]. However, the molecular regulatory mechanisms of sunflower in response to *S. sclerotiorum* remain poorly understood. In this study, we performed a leaf transcriptomic analysis of resistant and susceptible sunflower to determine the defense responses to *S. sclerotiorum* using RNA sequencing (RNA-seq). This work could provide new insight into understanding of the molecular mechanisms underlying the interaction between *S. sclerotiorum* and sunflower.

## Materials and methods

### Plant materials, inoculation and sampling

Both DR genotype C6 and DS genotype B728 are oilseed sunflower inbred lines which were bred by Liaoning Academy of Agricultural Sciences. The plants of these lines, which had similar growth periods, were cultivated in a growth chamber under a 16 h/8 h light/dark cycle (light intensity 4000 Lux) at 25°C. The infected inflorescences were sampled from field-grown sunflowers in Shenyang, Liaoning Province, China, to isolate and purify the *S. sclerotiorum*. The strain of *S. sclerotiorum* in present study was isolated and maintained on potato dextrose agar medium. A single mycelial plug (3 mm diameter) was cut from the culture to place at the

center of a new potato dextrose agar plate for incubation at 25˚C for 5 d in the dark. A total of ten mycelial agar plugs were collected from the edge of the colony for incubation in another new potato dextrose agar plate with the same conditions. Mycelial agar plugs were excised from the margin of a growing colony and used to inoculate sunflower leaves of 6 week old, while potato dextrose agar agar plugs without fungus was used as a mock-inoculated control. A randomized complete-block was used in the experimental design with three replications, and each replicate contained 20 plants. As the visible lesions appeared in DS genotype rather than DR genotype at 24 h post-inoculation (hpi), then the 20 leaves samples within the range of 0.5 cm bordering the extending lesion from inoculated and mock-inoculated plants were harvested for each replicate. Immediately after sampling, all samples were washed with sterilized water and placed into storage tubes, frozen with liquid nitrogen, and then stored at -80˚C for biochemical assays and RNA isolation.

## Microscopic observation

To observe the growth of *S. sclerotiorum* in sunflower, the leaves samples from inoculated plants of both genotypes were sampled at 24 hpi for wheat germ agglutinin (WGA) staining [22]. Single thin shavings were cut from the leaves samples, and placed in 2 ml microcentrifuge tubes. Then 10% KOH was used to soak the samples at room temperature, which had been rinsed once with distilled water. After rinsing three times with distilled water later, the samples were incubated with WGA stain solution, which consisted of 10 μg/ml WGA, 0.02% (v/v) Tween 20, 10 mM phosphate-buffered saline, pH 7.4. By using vacuum infiltration for 30 min, the samples were washed three times with phosphate-buffered saline then observed under a FV 1000 fluorescence microscope (OLYMPUS) with excitation at 488 nm and emission at 500 to 540 nm.

## Antioxidant enzyme extraction and activity assays

To obtain the enzymatic extract, frozen leaves (500 mg) from each sample was ground in liquid nitrogen to a fine powder. The tissue was resuspended in borate buffer to a final volume of 10 mL and centrifuged at 10,000 rpm for 15 min at 4˚C.

The superoxide dismutase activity (SOD) activity was determined as of the ability of photochemical reduction of nitrotetrazolium according to the method described by Giannopolitis and Ries [23]. The peroxidase (POD) activity was carried out by the method of Shannon et al [24]. The activity of catalase (CAT) was determined according to the method of Aebi [25]. The glutathione-S-transferase (GST) activity was measured as described by Mauch and Dudler [26]. The ascorbate peroxidase (APX) activity was obtained according to Nakano and Asada [27], using ascorbic acid as a substrate. The monodehydroascorbate reductase (MDAR) activity was assayed according to Drazkiewicz et al [28]. All the enzymatic analyses were performed in triplicate.

## RNA sequencing and analysis of RNA-seq data

Sunflower leaves were extracted using the RNAprep Pure Plant Kit (TIANGEN Biotech, Beijing, China) according to the instructions provided by the manufacturer. RNA samples were used to measure the concentration and purity in NanoDrop 2000 (Thermo Fisher Scientific, Wilmington, DE). The integrity of RNA samples was measured by using the RNA Nano 6000 Assay Kit of the Agilent Bioanalyzer 2100 system (Agilent Technologies, CA, USA). In this study, we used RNA samples with a 260/280 ratio of 1.96–2.17, a 28S/18S ratio of 1.0, and a RNA integrity number greater than 8.0 to prepare RNA-Seq libraries using NEBNext UltraTM RNA Library Prep Kit (NEB, USA) according to the manufacturer's instructions. The clustering of the index-coded samples was performed on a cBot Cluster Generation System using

TruSeq PE Cluster Kit v4-cBot-HS (Illumia). Finally, the RNA sequencing was performed on IlluminahpiiSeq 2500 platform by the Biomarker Technology, Qingdao, China.

Sequencing data were analyzed with the help of a bioinformatic pipeline tool, the BMKCloud platform (www.biocloud.net). Raw reads of fastq format were firstly processed through in-house perl scripts. In this step, the clean reads were obtained by removing adapter-containing reads, ploy-N-containing reads, and low quality reads from raw data. The Q20, Q30, GC-content and sequence duplication level of the clean data were calculated. All the downstream analyses were based on clean data with high quality. The adaptor sequences and low-quality sequence reads were removed from the data sets. Raw sequences were transformed into clean reads after data processing. These clean reads were then mapped to the reference genome sequence. Only reads with a perfect match or one mismatch were further analyzed and annotated based on the sunflower reference genome. The HanXRQr1.0 genome was used as reference in the present study [29]. Then, Hisat2 tools soft were used to map with reference genome. The StringTie Reference Annotation Based Transcript (RABT) assembly method was used to construct and identify both known and novel transcripts from Hisat2 alignment results. The gene expression levels are reflected by fragments per kilobase of transcript per million fragments mapped (FPKM). Genes with low expression (FPKM <1) were eliminated by default in the analysis process. An analysis of differential expression was carried out using the DESeq2 package. The false discovery rate (FDR) was controlled by using the Benjamini and Hochberg's approach to adjust the resulting P values. The genes were identified as DEGs when the FDR $\leqq$ 0.05 as well as the fold change $\geqq$ |2|.

## Functional classification and pathway analysis

Enrichment of Gene Ontology (GO) terms of the DEGs was implemented by AgriGO v2.0 (http://systemsbiology.cau.edu.cn/agri-GOv2/) with hypergeometric statistical test and Hocberg (FDR). The Kyoto Encyclopedia of Genes and Genome (KEGG, http://www.genome.jp/kegg/) was utilized for pathway enrichment analysis of DEGs. The overview of metabolic pathways in which DEGs involved in response to *S. sclerotiorum* were visualized by using MapMan software. In the MapMan software, the natural log of the ratio of the experimental mean to the control mean of the above called DEGs was taken in input files. Final analyses were performed with MapMan version 3.6.0 including the automatic application of the Wilcoxon rank sum test. Mapping files were exported from MapMan.

## Validation of RNA-seq analysis by qPCR

The extracted RNA was also used for qRT-PCR analysis to confirm the DEG results from RNA-seq. The same samples for RNA-seq were also used to extract the total RNA, which was further used to synthesize the cDNA. The qRT-PCR was conducted using ChamQ Universal SYBR qPCR Master Mix (Vazyme Biotech co., ltd, NanJing, China) on a Light Cycler 480 II (Roche, Basel, Switzerland). Primer sequences are described in S1 Table. The cycle threshold (CT) values obtained from three replicates and four technical replicates. The fold changes were calculated by $2^{-\Delta\Delta CT}$. $\Delta\Delta CT = (CT_{target} - CT_{action})$treatment—$(CT_{target} - CT_{action})_{control}$. Additionally, the results of qRT-PCR were presented as $\log_2$ fold changes to assess the accuracy of the RNA-seq.

## Results

### Symptoms and physiological traits in leaves of sunflower inoculated with *S. sclerotiorum*

Leaves from 6-week-old plants of two sunflower inbred lines, DR genotype C6 and DS genotype B728, were inoculated with isolated *S. sclerotiorum* using mycelial agar plugs. Visible

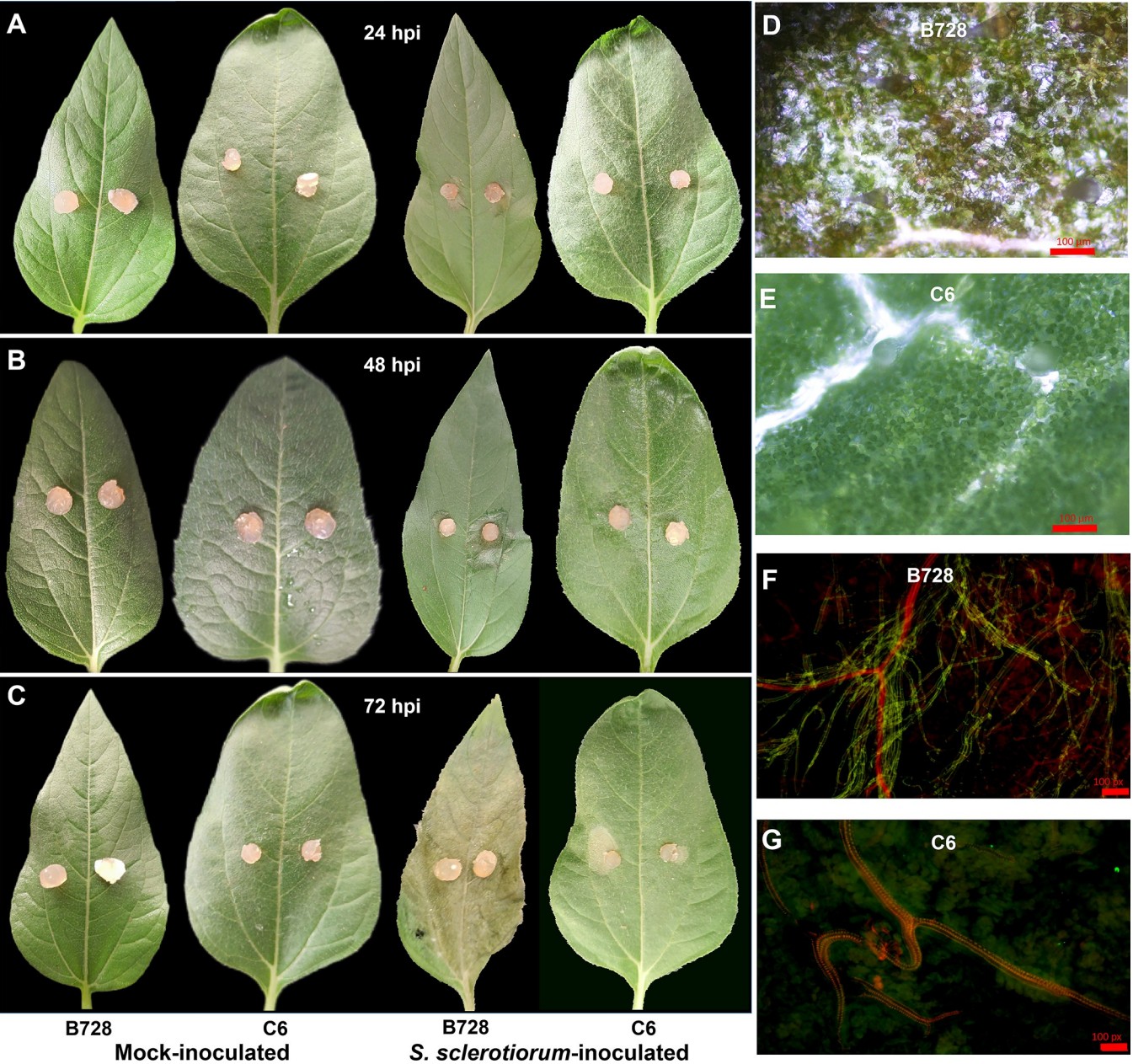

**Fig 1.** Visible lesions of leaves at 24 hpi (A), 48 hpi (B) and 72 hpi (C) and chlorosis phenomenon (D and E) and mycelial growth (F and G) of two genotypes at 24 hpi by electron microscopy. Fungal hyphae shows green fluorescence, cell walls of plant show red fluorescence (F and G) of two genotypes at 24 hpi by Leaf Fungal Hyphae WGA-PI staining.

lesions appeared in DS genotype at 24 hpi (Fig 1A) and in DR genotype at approximately 72 hpi (Fig 1C). The lesions developed more rapidly in DS genotype than in DR genotype. After 48 hpi (Fig 1B & 1C), the lesions on the leaves of DR genotype were significantly smaller than those on those of DS genotype. By electron microscopy, we found chlorosis in the leaf tissue of DS genotype (Fig 1D) but not on DR genotype at 24 hpi (Fig 1E). Microscopic assessments of the hyphae further showed that the pathogen was well established in the host tissue and led to extensive mycelial growth in DS genotype at 24 hpi (Fig 1F), while no visible growth of the hyphae was found in DR genotype (Fig 1G).

**Table 1. Activity changes in antioxidant enzymes in DS (B728) and DR (C6) sunflower inbred lines at 24 hpi by S. sclerotiorum infection.**

| Genotype | Treatment | SOD activity (ul·mg$^{-1}$ protein·min$^{-1}$) | POD activity (mmol·mg$^{-1}$ protein·min$^{-1}$) | CAT activity (mmol·mg$^{-1}$ protein·min$^{-1}$) | GST activity (μmol·mg$^{-1}$ protein·min$^{-1}$) | APX activity (mmol·mg$^{-1}$ protein·min$^{-1}$) | MDAR activity (mmol·mg$^{-1}$ protein·min$^{-1}$) |
|---|---|---|---|---|---|---|---|
| B728 | Control | 0.95±0.11b | 14.35±1.35d | 10.32±1.16d | 20.64±2.02d | 0.95±0.12a | 5.37±0.92b |
|  | inoculated | 1.14±0.12b | 24.36±2.02b | 25.34±2.33b | 31.27±2.33b | 0.68±0.13b | 3.28±0.84c |
| C6 | Control | 1.02±0.11b | 18.37±1.39c | 13.87±2.01c | 25.64±2.12c | 0.98±0.09a | 6.11±0.81a |
|  | inoculated | 2.15±0.15a | 30.11±2.87a | 29.72±2.56a | 35.69±2.24a | 1.02±0.12a | 6.59±0.72a |

The physiological response of leaves to *S. sclerotiorum* inoculation was further evaluated based on the activity of antioxidant enzymes, such as SOD, POD, CAT, GST, APX and MDAR. These data are shown in Table 1. Following infection, the activity of POD, CAT and GST were increased significantly in both genotype, while the SOD activity was increased significantly in DR genotype rather than DS genotype. APX and MDAR activity in DS genotype was decreased significantly by *S. sclerotiorum* infection, while DR genotype showed non-significant change. *S. sclerotiorum* infection, caused higher activity of the enzymes relative to the DS genotype.

## RNA-seq analysis and identification of DEGs

An average of 43 million raw reads were obtained for each sample, and 74.8–91.8% of trimmed reads were mapped to the reference genome. With a cutoff value of 1, a total of 22027 and 22579 genes were expressed in mock-inoculated samples of DS and DR genotypes, respectively, while 20711 and 23636 in the *S. sclerotiorum*-inoculated samples.

In order to determine whether DS and DR genotypes respond differently to *S. sclerotiorum* inoculation, DEGs were obtained by comparing gene expression levels between *S. sclerotiorum*- and mock-inoculated samples of the same genotype. In DS genotype, 10151 DEGs were identified between *S. sclerotiorum*- and mock-inoculated samples: 5481 up-regulated and 4670 down-regulated, whereas the same analysis in the DR genotype revealed only 7439 DEGs: 4221 up-regulated and 3218 down-regulated (Fig 2A). Venn diagrams showed that 2260 and 1000 genes were uniquely up-regulated in DS and DR genotypes, respectively, while 2557 and 1105 were uniquely down-regulated (Fig 2B). In addition, 119 genes were only induced in DS genotype after *S. sclerotiorum* infection but have constitutively expressed in the DR genotype (S2 Table).

## Gene ontology (GO) enrichment analysis of DEGs

AgriGO software was used to calculate the singular enrichment analysis (SEA) for the up- and down-regulated genes in DS genotypes, highlighted 126 and 134 GO significantly enriched terms, respectively (S3 and S4 Tables), while 98 and 88 GO terms were significantly enriched the up- and down-regulated genes in DR genotype (S5 and S6 Tables). We identified 249 and 282 common GO terms that were enriched in both the genotypes from up- and down-regulated genes in response to *S. sclerotiorum*, respectively (S7 and S8 Tables), which play an important role in the disease resistance of sunflower. For the up-regulated genes, 155 GO terms including protein kinase activity, protein autophosphorylation, response to oxidative stress, chitin catabolic process, immune response, cell surface receptor signaling pathway, nucleotide binding, plasma membrane and peroxidase activity were more enriched in DR genotype than in DS genotype (S7 Table). Among the 145 GO terms for the down-regulated genes, there were more enriched terms in DS genotype than in DR genotype. These included

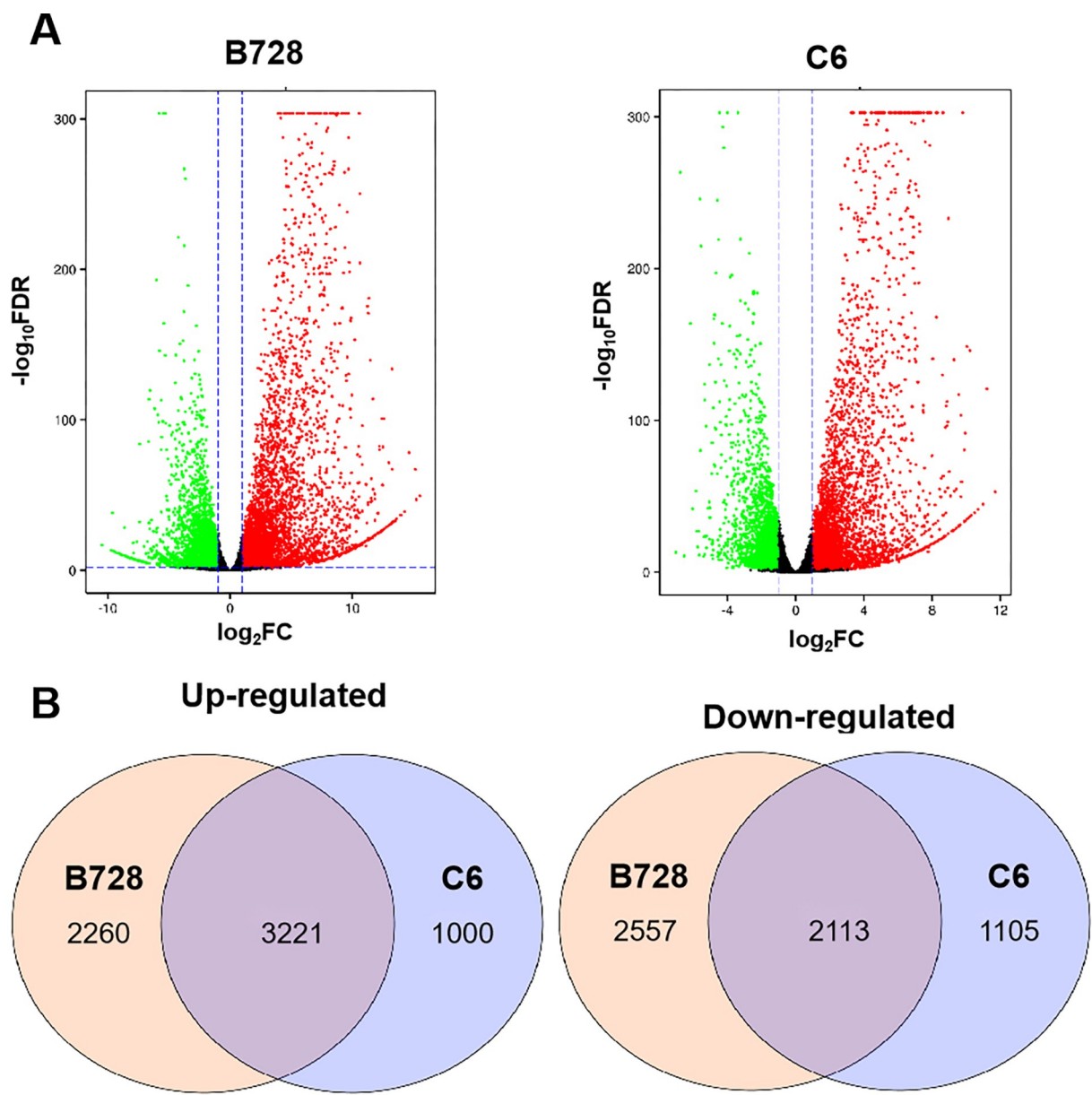

**Fig 2. Differentially expressed genes (DEGs) in each pairwise comparison.** (A) Volcano plot. The x-axis and y-axis represents the log2 (fold change) and −log10 (FDR-value). The red dots represent up-regulated genes with $log_2$ Fold_change >1 and FDR <0.01 and the green dots represent down-regulated genes with $log_2$ Fold_change < −1 and FDR <0.01. (B) Based on the two comparisons performed, two Venn diagrams showed the numbers of genes that are up- and down-regulated. Overlapping circles represent genes up- or down-regulated in both comparisons.

photorespiration, chlorophyll biosynthetic process, photosynthesis, chloroplast, plastid, thylakoid, organelle subcompartment, oxidoreductase activity and chlorophyll binding (S8 Table).

The SEA analysis was carried out on the uniquely up- and down-regulated genes in DS genotypes. Among the uniquely up-regulated genes 26 GO terms were significantly enriched: "intracellular protein transport" in the BP category, "clathrin vesicle coat" in CC category, and "GTPase activity" in MF category (S9 Table). "Carbohydrate metabolic process" in BP category, "vacuole" in CC category and "metal ion binding" in MF category were the most

**Table 2. Top 20 uniquely up-regulated genes in the DR genotype.**

| Gene | Log₂ FC | | Annotation |
|---|---|---|---|
| | **B728** | **C6** | |
| LOC110895548 | - | 9.24 | Protein PLANT CADMIUM RESISTANCE 2 |
| LOC110929546 | - | 8.93 | Expansin-like B1 |
| LOC110918750 | - | 8.25 | Lignin-forming anionic peroxidase |
| LOC110885989 | - | 8.21 | 1-aminocyclopropane-1-carboxylate oxidase 3 |
| LOC110886982 | - | 7.85 | Berberine bridge enzyme-like 13 |
| LOC110922271 | - | 7.83 | EP1-like glycoprotein 4 |
| LOC110879630 | - | 7.79 | Probable L-gulonolactone oxidase 6 |
| LOC110935776 | -1.62 | 7.78 | TMV resistance protein N |
| LOC110871630 | - | 7.73 | Regulator of G-protein signaling 1 |
| LOC110940063 | - | 7.70 | Dirigent protein 21 |
| LOC110928163 | - | 7.53 | Cytochrome P450 87A3 |
| LOC110877038 | - | 7.43 | Cytokinin dehydrogenase 3 |
| LOC110888580 | - | 7.39 | Probable carboxylesterase 18 |
| LOC110901085 | - | 7.38 | - |
| LOC110934894 | - | 7.32 | Ethylene-responsive transcription factor ERF098 |
| LOC110925525 | - | 7.29 | Hydroxycinnamoyltransferase 4 |
| LOC110909848 | - | 7.25 | Short-chain dehydrogenase reductase 2a |
| LOC110899788 | - | 7.21 | 1-aminocyclopropane-1-carboxylate oxidase homolog 1 |
| LOC110945103 | - | 7.13 | Cytochrome P450 71AV8 |
| LOC110900121 | - | 7.06 | Hydroquinone glucosyltransferase |

enriched GO terms among the 118 identified from unique down-regulated genes in the DS genotype (S10 Table).

The SEA analysis was also carried out on the uniquely up- and down-regulated genes in DR genotypes. Among the uniquely up-regulated genes 63 GO terms were significantly enriched: "cellular polysaccharide metabolic process" in the BP category, "cellular polysaccharide biosynthetic process" in CC category, and "cell morphogenesis" in MF category (S11 Table). "Cellular component organization" in BP category, "plasma membrane" in CC category and "transcription factor activity, sequence-specific DNA binding" in MF category were the most enriched GO terms among the 16 identified from unique down-regulated genes in the DR genotype (S12 Table).

Top twenty uniquely up- (log₂FC >7) or down-regulated (log₂FC <-4) genes were identified in the DR genotypes, respectively (Tables 2 and 3), most of which were enriched in the above uniquely up- or down-regulated GO terms. These uniquely up-regulated genes in the DR genotype were annotated as cytokinin dehydrogenase, cytochrome P450, expansin, transcription factor ERF, TMV resistance protein, etc (Table 2). These uniquely down-regulated genes in DS genotype were annotated as proline-rich protein, dehydration-responsive element-binding protein, leucine-rich repeat receptor protein kinase, bidirectional sugar transporter SWEET14, *etc* (Table 3).

## Pathway analysis of DEGs

KEGG pathway analysis was performed on the DEGs in response to *S. sclerotiorum* infection in DS (S1 Fig) and DR (S2 Fig) genotypes. The plant-pathogen interaction was the top up-regulated pathway for both genotypes. In this pathway, 24 and 23 receptor-like proteins, which could be considered as PRRs, were up-regulated significantly by *S. sclerotiorum* infection in

**Table 3. Top 20 uniquely down-regulated genes in the DR genotype.**

| Gene | Log$_2$ FC | | Annotation |
|---|---|---|---|
| | **B728** | **C6** | |
| LOC110885111 | -5.70 | -3.21 | Proline-rich protein 4 |
| LOC110930392 | -5.55 | - | Early light-induced protein 2 |
| LOC110891330 | -5.40 | - | Soyasapogenol B glucuronide galactosyltransferase |
| LOC110869264 | -4.92 | - | Dehydration-responsive element-binding protein 1D |
| LOC110921230 | -4.85 | -2.19 | - |
| LOC110909326 | -4.54 | - | - |
| LOC110869165 | -4.54 | - | Cyclin-U4-1 |
| LOC110915744 | -4.54 | - | Leucine-rich repeat receptor protein kinase HPCA1 |
| LOC110941770 | -4.49 | - | Bidirectional sugar transporter SWEET14 |
| LOC110895020 | -4.40 | 0.33 | Probable galacturonosyltransferase-like 9 |
| LOC110917579 | -4.37 | -0.67 | Protein PHOSPHATE-INDUCED 1 |
| LOC110908033 | -4.19 | -2.38 | UPF0481 protein At3g47200 |
| LOC110928088 | -4.15 | - | Protein SRC2 |
| LOC110873198 | -4.09 | -1.73 | Transcription repressor OFP1 |
| LOC110929642 | -4.06 | -1.37 | Receptor-like protein kinase THESEUS 1 |
| LOC110912169 | -4.06 | -3.72 | Probable inactive receptor kinase At1g48480 |
| LOC110931795 | -4.04 | -2.49 | - |
| LOC110864816 | -4.01 | - | GDSL esterase/lipase At4g01130 |
| LOC110929231 | -3.98 | -1.20 | Calmodulin-binding receptor kinase CaMRLK |
| LOC110868988 | -3.94 | -2.02 | Uncharacterized protein At5g39865 |

DS and DR genotypes, respectively (Table 4). Of these, 13 were uniquely up-regulated in DR genotype and its expression exhibited non-significantly changed in DS genotype, such as EIX1 and EIX2 homolog genes.

By using MapMan software, a pathway was shown consisting of 1089 and 886 DEGs by *S. sclerotiorum* infection in DS and DR genotypes (Fig 3). In both genotypes, most of genes involved in effector receptor (NLR), protein kinase MAPK, calcium-dependent signaling (*e.g.* CDPKs, CBLs and CMLs), transcription factor (*e.g.* ERF, WRKY and MYB), systemic acquired resistance (*e.g.* regulatory protein CBP60/SARD, disease resistance mediator MLO2/6/12, aminotransferase ALD1, pipecolate N-hydroxylase FMO1 and pipecolate oxidase SOX) and secondary metabolism (*e.g.* mevalonate pathway, methylerythritol phosphate pathway, p-coumaroyl-CoA biosynthesis and flavonoid biosynthesis) were up-regulated, while most genes involved in proteolysis (*e.g.* cysteine-type peptidase, serine-type peptidase, aspartic-type peptidase and metallopeptidase activities) were down-regulated by *S. sclerotiorum* infection. Of above pathways, the number of up- or down-regulated genes in DS genotype was more than in DR genotype. However, a large number DEGs were also divided into hormone signaling, cell wall organisation and redox homeostasis. There were chosen for deeper evaluation of *S. sclerotiorum* resistant traits.

**Plant hormone signaling pathways.** Most auxin signaling pathways were significantly down-regulated following infection. The number of these down-regulated genes were more in DS genotype than in DR genotype (S13 Table). However, most of genes related to signaling pathways of ethene (ET) and brassinolide (BR), were significantly up-regulated. The DS genotype had more up-regulated genes than the DR genotype in above pathways apart from ET signaling. The ET signaling pathways showed more up-regulated genes in DR genotype (11) compared to DS genotype (7), which was attributed to more uniquely up-regulated genes for

**Table 4. A list of transmembrane recognition receptors (PRRs) involved in plant-pathogen interaction pathway by KEGG analysis.**

| Gene | Log$_2$ FC | | Annotation |
|---|---|---|---|
| | B728 | C6 | |
| LOC110867023 | 5.60 | 6.46 | Receptor-like protein 12 |
| LOC110883941 | 4.02 | 3.60 | Receptor-like protein 12 |
| LOC110905009 | 2.29 | 1.71 | Receptor-like protein EIX1 |
| LOC110943528 | 2.50 | 1.42 | Receptor-like protein EIX1 |
| LOC110907898 | 5.24 | 3.84 | Receptor-like protein EIX1 |
| LOC110867197 | 1.66 | 1.20 | Receptor-like protein EIX2 |
| LOC110894990 | 6.67 | 4.51 | Receptor-like protein EIX2 |
| LOC110901887 | 3.27 | 2.31 | Receptor-like protein EIX2 |
| LOC110884455 | 5.47 | 3.79 | Receptor-like protein EIX2 isoform X1 |
| LOC110904904 | 3.57 | 1.12 | Receptor-like protein EIX2 isoform X1 |
| LOC110911286 | -1.60 | -1.26 | Receptor-like protein 12 |
| LOC110927258 | 7.29 | - | Receptor-like protein 12 |
| LOC110880537 | 5.50 | - | Receptor-like protein 12 |
| LOC110898370 | 2.33 | - | Receptor-like protein 12 |
| LOC110899583 | 6.89 | - | Receptor-like protein 32 |
| LOC110932987 | 7.90 | - | Receptor-like protein 37 |
| LOC110911343 | 2.13 | - | Receptor-like protein EIX1 |
| LOC110935701 | 2.81 | - | Receptor-like protein EIX1 precursor |
| LOC110883962 | 2.01 | - | Receptor-like protein EIX2 |
| LOC110899544 | 3.10 | - | Receptor-like protein EIX2 |
| LOC110932925 | 7.28 | - | Receptor-like protein EIX2 |
| LOC110932954 | 6.64 | - | Receptor-like protein EIX2 |
| LOC110909891 | 1.04 | - | Receptor-like protein EIX2 |
| LOC110883961 | 2.45 | - | Receptor-like protein EIX2 |
| LOC110877393 | 1.16 | - | Receptor-like protein EIX2 isoform X1 |
| LOC110873395 | -1.82 | - | Receptor-like protein EIX2 |
| LOC110880683 | - | 2.01 | Receptor-like protein 12 |
| LOC110904906 | - | 1.07 | Receptor-like protein 12 |
| LOC110872328 | - | 1.03 | Receptor-like protein 12 |
| LOC110867178 | - | 1.33 | Receptor-like protein EIX1 |
| LOC110872190 | - | 1.72 | Receptor-like protein EIX2 |
| LOC110932995 | - | 1.43 | Receptor-like protein EIX2 |
| LOC110927229 | - | 1.11 | Receptor-like protein EIX2 |
| LOC110924352 | - | 1.79 | Receptor-like protein EIX2 |
| LOC110887861 | - | 1.55 | Receptor-like protein EIX2 |
| LOC110880528 | - | 3.40 | Receptor-like protein EIX2 isoform X1 |
| LOC110867209 | - | 1.85 | Receptor-like protein EIX2 isoform X1 |
| LOC110867212 | - | 1.40 | Receptor-like protein EIX2 isoform X1 |
| LOC110872808 | - | 1.09 | Receptor-like protein EIX2 isoform X1 |

receptor protein (ETR/ERS) and EIN3 transcription factor activity (Fig 4A). After *S. sclerotiorum* infection, only the gene LOC110884479 encoding EIN3 transcription factor activity showed higher expression level in DR genotype than in DS genotype.

**Redox homeostasis.** Redox homeostasis was affected by *S. sclerotiorum* infection, in which 94 DEGs in DS genotype and 68 in DR genotype. Most of genes for NADPH respiratory burst oxidase homolog involved in the ROS generation were up-regulated in both genotypes

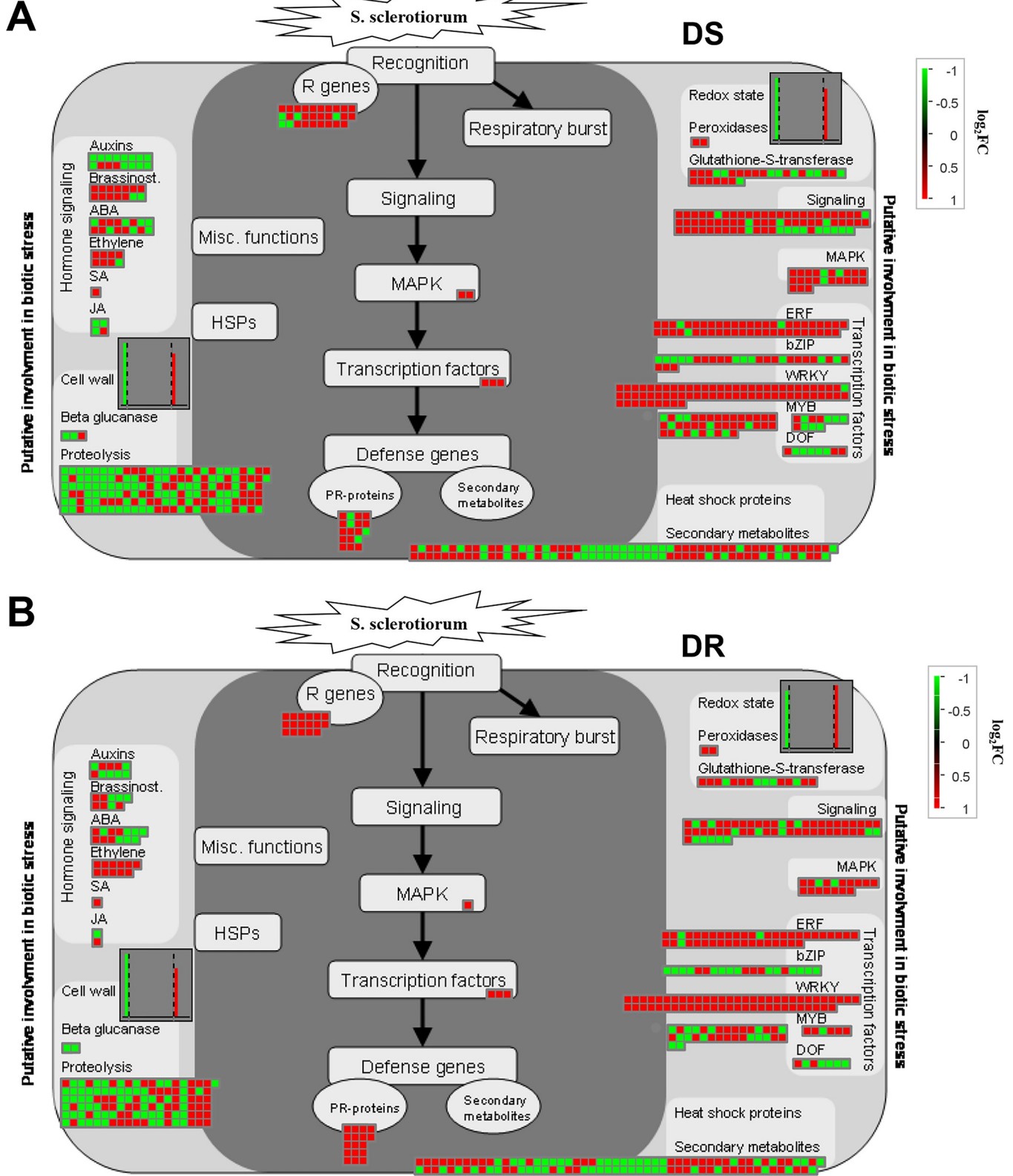

**Fig 3.** Infection with *S. sclerotiorum* results in DEGs at the DS (A) and DR (B) genotypes as visualized in MapMan. Red and green indicate DEGs that are up-regulated and down-regulated, respectively.

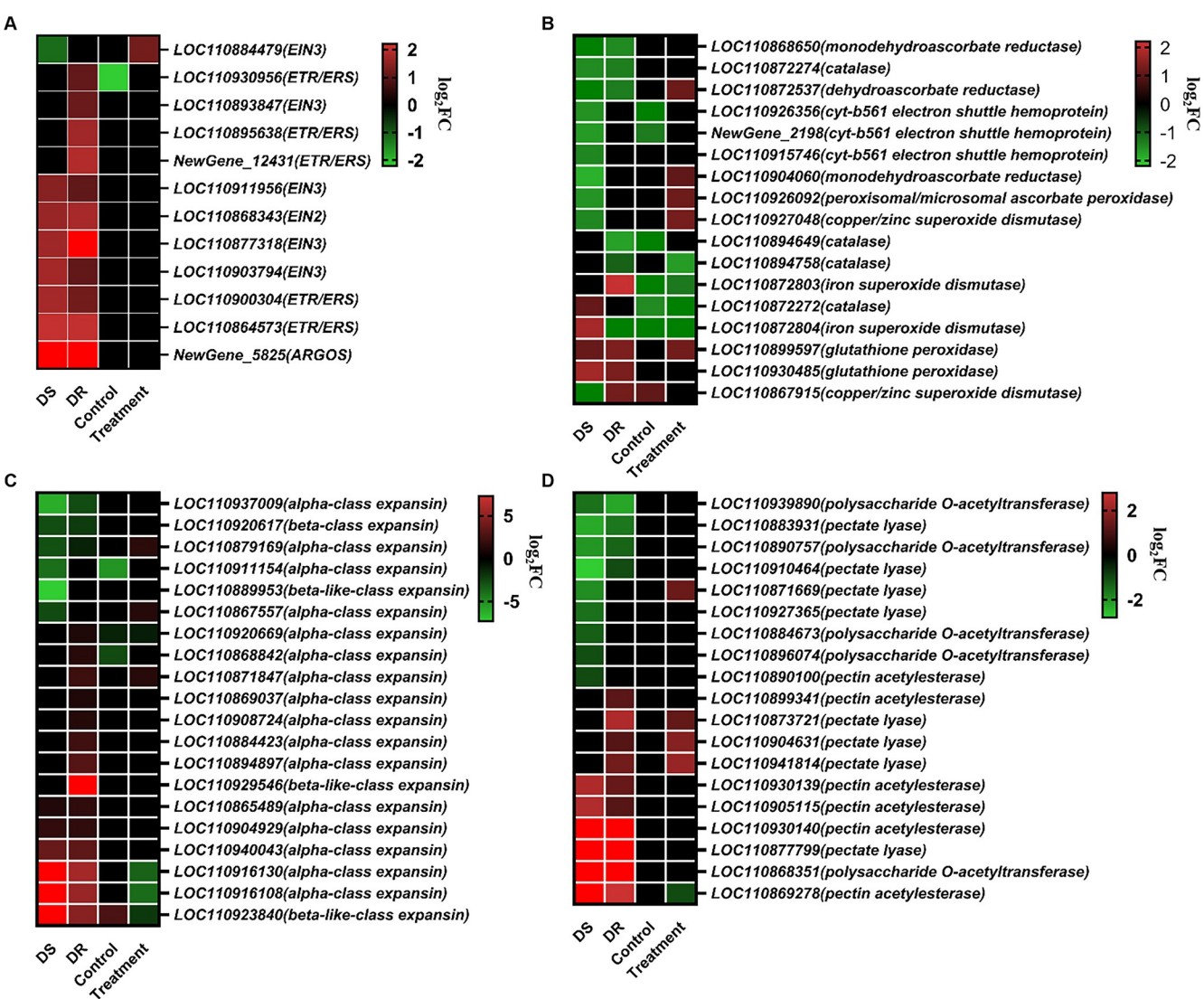

**Fig 4.** Heat map of DEGs related to ET signaling (A), ROS (B), expansin (C) and pectin (D). The gene expression levels were compared between *S. sclerotiorum*- and mock-inoculated samples of the same genotype as well as between the DR and DS genotype under the same conditions.

(S14 Table). In ROS scavenging, the DR genotype showed more up-regulated genes associated with SODs than the DS genotype (Fig 4B). On the contrary, the DS genotype had more down-regulated genes involved in ascorbate-based redox regulation including APX and MDAR than the DR genotype. After *S. sclerotiorum* infection, the DR genotype showed six up-regulated and four down-regulated genes as compared to DS genotype.

**Cell wall organization.** In cell wall organisation, 285 and 262 DEGs were identified by *S. sclerotiorum* infection in DS and DR lines, respectively. These were involved in the cellulose synthase complex, xylan biosynthesis, heteromannan biosynthesis, and cutin polyester biosynthesis were down-regulated in both genotypes, while monolignol biosynthesis and expansin activities were up-regulated (S15 Table). However, we found several up-regulated genes related to expansin activities, such as alpha-class expansin, which were more in the DR genotype (Fig 4C). In addition, genes involved in pectin modification and degradation, such as pectate lyase

showed a peculiar behaviour, in that most of them were up-regulated in the DR genotype rather than the DS one (Fig 4E). Following *S. sclerotiorum* infection, four genes LOC110871669, LOC110873721, LOC110904631 and LOC110941814 encoding pectate lyase showed higher expression level in DR genotype than in DS genotype.

### Validating the reliability of RNA-seq data by qRT-PCR

Nine genes were selected to validate the reliability of RNA-seq data by using qRT-PCR. The expression trends of selected genes obtained by qRT-PCR were in agreement with those measured by RNA-seq (Fig 5). Of these, the CDPK9 (LOC110873588) and MAPK5 (LOC110935435) were significantly up-regulated by *S. sclerotiorum* infection in both genotypes (Fig 5A and 5B). The CML23 (LOC110869082) and five NLRs (LOC110915177, LOC110877402, LOC110923780, LOC110883529 and LOC110929730) were uniquely up-regulated in the DR genotype and their expression but showed non-significantly changed in DS genotype (Fig 5C–5H). Notably, the LOC110935776 encoding NLR exhibited a the opposite expression pattern, which was down-regulated in DS genotype but dramatically up-regulated in the DR genotype (Fig 5I).

## Discussion

### Differences in *S. sclerotiorum* resistance between the two sunflower genotypes

In this study, the DR genotype C6 displayed symptoms of *S. sclerotiorum* infection later (72 hpi) and with less severity after inoculation compared to the DS genotype B728 (24 hpi). This was also reported by previous studies in regard to the defence response to *S. sclerotiorum* in other crop plants [30,31]. The assessments of visible lesions and hyphae further demonstrated the resistance difference between these two genotypes. We sampled the inoculated and mock-inoculated leaves at 24 hpi for RNA-seq analysis, when the slight phenotypic difference has just occurred. At this time point, the DS genotype had larger number of DEGs than the DR genotype, indicating that it is relatively hyper-responsive or more sensitive to *S. sclerotiorum* at the molecular level. The results were inconsistent with the results of previous studies in B. *napus* [30,32].

### Expansin and pectate lyase activities contribute to resistance against *S. sclerotiorum*

We identified more down-regulated genes involved in chloroplast as well as photosynthesis in DS genotype compared with those in the DR genotype. Disruption of chloroplast function often resulted in the differentially expressed of genes involved in cell wall synthesis and modification. Cell wall, as the first defensive barrier against pathogens, undergoes dynamic remodeling to prevent infection by pathogens [33]. During the interaction with its host, *S. sclerotiorum* synthesizes and secretes oxalic acid and cell-wall-degrading enzymes, which could decrease cell wall rigidity, cause rapid cell collapse, and finally destroy expanding parts of infected tissue [34]. The extracellular enzymes produced by *S. sclerotiorum* can physically degrade plant cell wall components and consequently provide nutrients to drive the infection process [35]. Up-regulation of cellulose synthases, xyloglucan endotransglucosylase/hydrolase, monolignol biosynthesis, lignin deposition could reinforce the cell wall to react faster and stronger to subsequent *S. sclerotiorum* attack [36–38]. In this study, most of genes related to monolignol biosynthesis and expansin activities were up-regulated by *S. sclerotiorum* infection in both genotypes. Importantly, we found more up-regulated genes related to expansin activities in

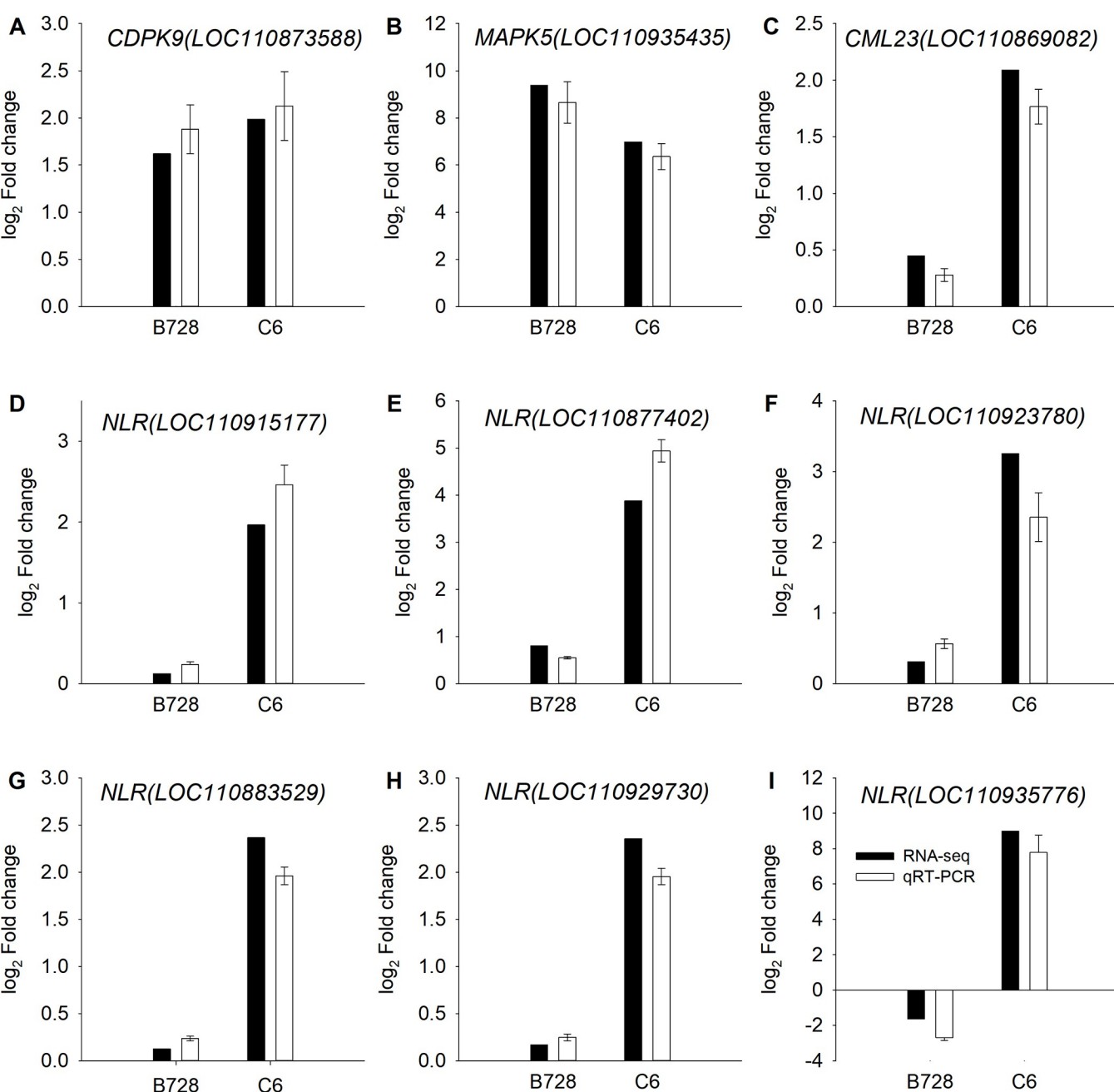

**Fig 5. The expression pattern of nine genes were verified by qRT-PCR.**

DR genotype compared to DS genotype. Plant expansins as cell wall protein play a key role in many processes modify the cell wall, including relaxation and extension [39], and constitute a regular component against pathogen infection [40]. Overexpression of *EXLB8*, one of subfamilies of plant expansins, improved resistance to *S. sclerotiorum* in tobacco through physically modifying the cell wall structure as well as activating the phytohormone signaling pathways, such as JA-ET cooperation [41].

In addition, we found that the genes for pectate lyase were up-regulated in DR genotype rather than DS genotype by *S. sclerotiorum* infection. The homogalacturonan of pectin that

could expose the polysaccharides at the early stage of invasion, was depolymerized by pectate lyase, which was released from pathogens [42–44]. By the degradation of polysaccharides, the plant cell wall produce oligosaccharides, which can enhance disease resistance via promoting ROS burst, MAPK activation and thus the expression of immune-responsive genes [45,46]. Our results suggested that the expansin and pectate lyase activities may be associated with *Sclerotinia* resistance in sunflower.

## Differences in ROS scavenging between contrasting genotypes

The ROS are secondary messengers in the signal transduction process and play crucial roles in stress response and plant growth [47]. However, if the stress is too severe, excess ROS will cause programmed cell death and trigger host cell necrosis, which can promote the growth of necrotrophic pathogens such as *S. sclerotiorum* [48]. Organisms have complex antioxidant systems that regulate the balance between ROS production and ROS scavenging to combat oxidative stress caused by excess ROS [49]. SOD is involved in inactivating superoxide anions, which constitutes the front-line defense against ROS [50]. In this study, the increase of SOD activity was found only in the DR genotype after *S. sclerotiorum* infection. Most of genes associated with SOD were up-regulated in the DR genotype, indicated an elevated levels of antioxidant defence decreased oxidative damage and cell death in this genotype. Moreover, the pathway of ascorbate-glutathione cycle is essential for ROS scavenging, which can protect ROS signalling and avoid $H_2O_2$ toxicity [51,52]. We found higher activity of GST, APX and MDAR in DR genotype than in DS genotype after *S. sclerotiorum* infection. GSTs play a crucial role as cellular protectants in detoxification against oxidative damage, and were induced by *S. sclerotiorum* in this study, which has been also reported in the previous studies [53]. However, the DS genotype had more down-regulated genes associated with APX and MDAR involved in ascorbate-based redox regulation. APX as one of the most important peroxidases catalyzes reduction of $H_2O_2$ by the reducing power of ascorbate [54], which is the most important reducing substrate for $H_2O_2$ detoxification in plant cells [55]. MDAR can directly catalyses the regeneration of ascorbate at both plasmalemma and thylakoid membrane [56]. During the detoxification of ROS, Additionally, MDAR is capable of converting phenoxyl radicals into their respective parental phenols, which are very efficient antioxidants comparable to ascorbate [57]. Decreased activity of APX and MDAR only in DS leaves likely contributes to the cellular degradation before fungal hyphae, thereby increasing nutrient availability after OA-induced cell death [58].

## Immune response to *S. sclerotiorum* in sunflower

The initial stage of PTI involves PRR stimulation [58]. A total of 57 genes for receptor-like kinases (RLKs) were as well-characterized PRR, which were up-regulated in *Brassica napus* by *S. sclerotiorum* infection [30]. A PRR gene, *ZmWAK*, was identified in maize by QTL mapping, which was up-regulated after infected and confered quantitative resistance to head smut [57]. In this study, 24 and 23 genes, in the DR and DS genotype, respectively, were identified as PRRs in the plant-pathogen interaction pathway, which were up-regulated by *S. sclerotiorum* infection. Importantly, we found 13 PRRs were uniquely up-regulated in the DR genotype, which might have a critical role in recognizing pathogen-associated molecular patterns induced by S. sclerotiorum and initiating the immune response.

   Defense responses also begin with the intracellular recognition of the pathogen effector molecules through a series of plant R gene products, which leads to ETI [58]. In disease resistance breeding, ETI as an important trait is widely used to resist the biotrophic and hemibiotrophic instead of necrotrophic pathogens [59]. However, in *Arabidopsis*, RLM3 has been

identified as R gene, which encodes a TIR domain known to affect broad-range immunity to several necrotrophic pathogens [60]. It is unknown whether sunflower has any R-genes may refered to the *S. sclerotiorum* resistance. In this study, we identified six R genes encoding NLR proteins by qRT-PCR, which were uniquely up-regulated in the DR genotype. Of these, the gene LOC110935776 was down-regulated in DS genotype but dramatically up-regulated in DR genotype. In sunflower, it will be necessary to investigate whether these NLR genes play a role in the immune response to *S. sclerotiorum*.

### Defense signaling pathways in response to *S. sclerotiorum* challenge

When recognizing invading pathogens, plants respond to pathogen attack by activation a large number of defense responses, in which a central role is signaling, including MAPK kinases, calcium-dependent signaling, plant hormone and transcription factors. Most of genes associated with MAPK kinases, CDPKs, CMLs, ET, BR, ERF, WRKY and MYB were up-regulated by *S. sclerotiorum* infection in this study. During *S. sclerotiorum* infection, MAPK is an important serine/threonine protein kinase that transduces signals from cytoplasm to nucleus [61]. The changes of transcription factors dependent on a series of phosphorylation was due to the MAPK cascade signalling pathway, which were activated after PRR stimulation [62,63]. The transcription factors MPK4 and the WRKY33 were demonstrated to assoiated with the resistance to *Botrytis cinerea* and *S. sclerotiorum* [64–66]. Over-expression of BnMAPK4 enhances resistance to *S. sclerotiorum* in transgenic *Brassica napus* [67].

In addition, the plant respond to *S. sclerotiorum* attack was also depend on the calcium (Ca) signaling pathway, which mainly included CDPKs and CMLs. It is a well known that the $H_2O_2$ accumulation can be positively regulated by CDPKs via phosphorylating in the plasma membrane and cytosol, and thus activate RBOHD/F [68]. These results suggest that these genes as the regulator might play a key role in cellular signaling networks to induce *S. sclerotinia* stress resistance. In this study, one gene CML23 were identified as uniquely up-regulation in the DR rather than in DS genotype.

Jasmonic acid (JA)/ET and SA-mediated signaling pathways are known to be major components of plant defense networks [69,70], which can act independently, synergistically, or in some cases antagonistically between host/pathogens [71]. In this study, genes associated with ET signal pathways were induced, whilst no salicylic acid (SA) and JA responsive genes were identified. ET could induce the pathogenesis related proteins and synthesize the phytoalexin against invading microbial pathogens [72]. Previous study showed that the ET responsive signaling seemed to proceed at a late stage of disease in the tolerant *B. carinata* but no such increase was found in the susceptible ones at any of the investigated time points [73]. We also found that the DR genotype had more up-regulated genes associated with ETR/ERS and transcription factor EIN3 in the ET signaling pathways compared to DS genotype. Our results suggested that the increase of ET signal transduction could contribute to resistance against *S. sclerotiorum*. Oppositely, more up-regulated genes for ET biosynthesis were found in DS genotype than in DR genotype (S13 Table). ET has been demonstrated to play a key role in the programmed cell death regulation [74], which might help the necrotrophic pathogen to absorb nutrients for continued growth and development [70]. Thus, we proposed that blocking ET biosynthesis may alleviate *S. sclerotinia* in sunflower.

### Conclusions

RNA-Seq analysis was carried to analyze transcriptomic profiles of two sunflower genotypes viz., DS and DR that were systemically infected with *S. sclerotinia* and elucidate the molecular basis of the *S. sclerotiorum* resistance mechanism. The DEGs analysis showed that the DR

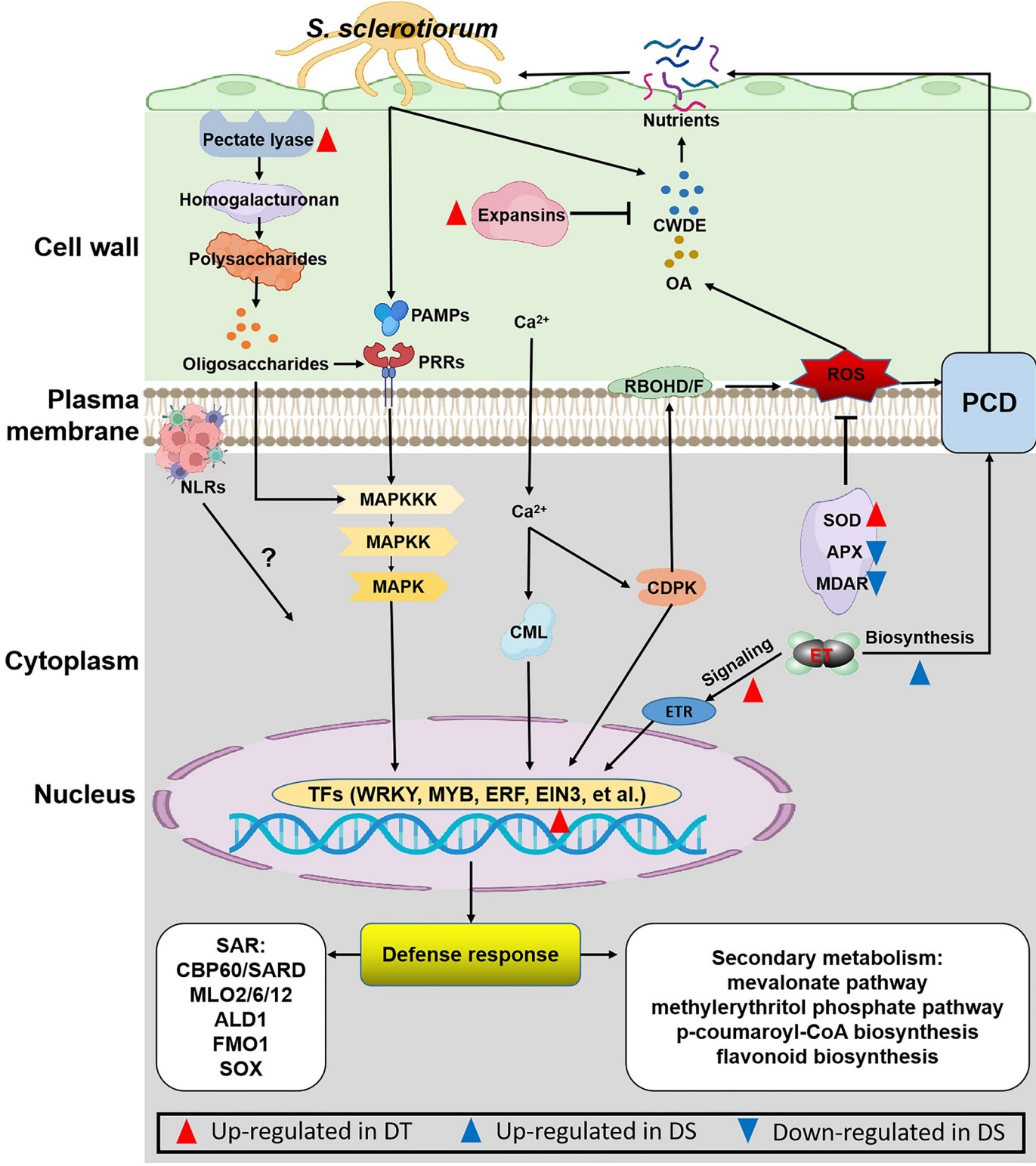

**Fig 6. Sunflower defense responses against *S. sclerotiorum* described in a molecular model.** CWDE: Cell wall-degrading enzymes; OA: Oxalic acid; TF: Transcription factor; MAPK- mitogen activated protein kinase.

genotype is less affected by *S. sclerotiorum* infection as compared to the DS genotype. Taken together the results from the GO, KEGG and MapMan analysis, a model was proposed to show the major molecular mechanisms underlying the defense responses against *S. sclerotiorum* in sunflower (Fig 6). The oxalic acid and cell-wall-degrading enzymes as well as the excess of ROS, generated by the interaction between host and *S. sclerotiorum*, cause the programmed cell death, which help the *S. sclerotiorum* to obtain nutrients for growth and development. In the DR genotype, the expansins were up-regulated to restrain cell-wall-degrading enzymes; pectate lyase activities were up-regulated to activate the defense response; the SOD was up-regulated to eliminate of ROS; the ET signaling were up-regulated to activate the systemic acquired resistance and secondary metabolisms via a series of signal transduction. In DS genotype, however, the up-regulated of ET biosynthesis and the down-regulated of APX and MDAR facilitated the programmed cell death. Our results revealed how the cell wall modifying enzymes, antioxidants, hormone as well as signaling increased *S. sclerotiorum* resistance in DR relative to DS sunflower genotype. Overall, these finding will serve as a resource for further studies of the molecular mechanisms associated the *S. sclerotiorum* resistance in sunflower and development of effective strategies in *Sclerotinia*-resistance breeding.

## Supporting information

**S1 Table. The gene-specific primers for qRT-PCR.**
(XLSX)

**S2 Table. 119 constitutive expressed genes are induced by *S. sclerotiorum* infection in the DS genotype and are already expressed in the DR genotype in mock-inoculated control.**
(XLSX)

**S3 Table. GO analysis to identify enriched gene ontologies among the up-regulated DEGs in DS genotype B728.**
(XLSX)

**S4 Table. GO analysis to identify enriched gene ontologies among the down-regulated DEGs in DS genotype B728.**
(XLSX)

**S5 Table. GO analysis to identify enriched gene ontologies among the up-regulated DEGs in DR genotype C6.**
(XLSX)

**S6 Table. GO analysis to identify enriched gene ontologies among the down-regulated DEGs in DR genotype C6.**
(XLSX)

**S7 Table. GO analysis to identify enriched gene ontologies among the common up-regulated DEGs in DS and DR genotypes.**
(XLSX)

**S8 Table. GO analysis to identify enriched gene ontologies among the common up-regulated DEGs in DS and DR genotypes.**
(XLSX)

**S9 Table. GO analysis to identify enriched gene ontologies among the unique up-regulated DEGs in DS genotype B728.**
(XLSX)

**S10 Table. GO analysis to identify enriched gene ontologies among the unique down-regulated DEGs in DS genotype B728.**
(XLSX)

**S11 Table. GO analysis to identify enriched gene ontologies among the unique up-regulated DEGs in DT genotype C6.**
(XLSX)

**S12 Table. GO analysis to identify enriched gene ontologies among the unique down-regulated DEGs in DT genotype C6.**
(XLSX)

**S13 Table. DEGs involved in phytohormone action by MapMan analysis.**
(XLSX)

**S14 Table. DEGs involved in redox homeostasis by MapMan analysis.**
(XLSX)

**S15 Table. DEGs involved in cell wall organisation by MapMan analysis.**
(XLSX)

**S1 Fig.** Kyoto Encyclopedia of Genes and Genome (KEGG) analysis for up- (A) and down-regulated (B) DEGs in DS genotype, respectively.
(TIF)

**S2 Fig.** Kyoto Encyclopedia of Genes and Genome (KEGG) analysis for up- (A) and down-regulated (B) DEGs in DR genotype, respectively.
(TIF)

## Author Contributions

**Data curation:** Mingzhu Zhao.

**Formal analysis:** Mingzhu Zhao.

**Investigation:** Mingzhu Zhao, Bing Yi, Xiaohong Liu, Enyu Sun.

**Methodology:** Mingzhu Zhao.

**Resources:** Dexing Wang, Dianxiu Song, Liangji Cui.

**Software:** Mingzhu Zhao.

**Validation:** Jingang Liu.

**Writing – original draft:** Mingzhu Zhao.

**Writing – review & editing:** Jingang Liu, Liangshan Feng.

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
