## [Decision Letter · Decision Letter 0]

2 Aug 2023

PONE-D-23-14214Comparative transcriptome analysis reveal the Sclerotinia sclerotiorum  resistance mechanism in sunflowerPLOS ONE

Dear Dr. Liu,

Thank you for submitting your manuscript to PLOS ONE. After careful consideration, we feel that it has merit but does not fully meet PLOS ONE’s publication criteria as it currently stands. Therefore, we invite you to submit a revised version of the manuscript that addresses the points raised during the review process.

Upon careful evaluation of the Reviewer(s)' comments and my own assessment, it is evident that significant revisions are necessary for your manuscript before it can be further considered for publication. I would like to inform you that I reserve the right to send the revised manuscript to the original Reviewer(s) or seek input from new Reviewer(s) to ensure a thorough evaluation before making a final decision on its suitability for publication.The comments of the Reviewer(s) are included at the bottom of this letter.

Main concerns

Unavailability of raw RNA-seq data

General findings

Lack of comprehensive analysis

Insufficient detail in data processing methods

Missing critical genome assembly information

We look forward to receiving your revised manuscript.

Kind regards,

Asela J. Wijeratne, Ph.D.

Academic Editor

PLOS ONE

Journal Requirements:

The study was supported by China Agriculture Research System of MOF and MARA (CARS-14), and Shenyang Seed Industry Innovation Project (22-318-2-19). 

Reviewers' comments:

Reviewer's Responses to Questions

**Comments to the Author**

1. Is the manuscript technically sound, and do the data support the conclusions?

Reviewer #1: Yes

Reviewer #2: Partly

2. Has the statistical analysis been performed appropriately and rigorously? 

Reviewer #1: Yes

Reviewer #2: No

3. Have the authors made all data underlying the findings in their manuscript fully available?

Reviewer #1: Yes

Reviewer #2: No

4. Is the manuscript presented in an intelligible fashion and written in standard English?

Reviewer #1: No

Reviewer #2: Yes

5. Review Comments to the Author

Reviewer #1: The paper provides good insights into transcriptional changes occurring in susceptible and resistant sun flower following Sclerotinia sclerotiorum infection. There is a good attempt to measure physiological and correlate it with infection. The authors also provide a good model and graphical abstract as figure 8. For this, I commend the authors.

Having said that, my biggest sticking point with the paper is the poor writing. It was difficult to understand some parts of the paper because of the way poor English and grammar. I pointed out a few and suggested revisions, but the whole paper will need to go through a thorough editing process.

I was also less impressed by the quality of figures 4 and 5.

Below are my specific comments:

Line 3

S. sclerotiorum

Page 4 line 2

Delete “thus far”

Page 4 line 18-19

Rephrase as follows:

“Both DR genotype C6 and DS genotype B728 are oilseed sunflower inbred lines which were bred by Liaoning Academy of Agricultural Sciences”

Line 23

Italicize “S. Sclerotiorum”

Line 5 Page 5

Replaced “were washing” with “washed”

Page 7 line 2 and throughout the paper,

Italicize “S. sclerotiorum”

Page 8 lines 3-12

Consider rephrasing as follows:

“The physiological response of leaves to S. sclerotiorum inoculation was further evaluated based on the activity of antioxidant enzymes, such as superoxide dismutase activity (SOD), peroxidase (POD), catalase (CAT), glutathione-S-transferase (GST), ascorbate peroxidase (APX) and monodehydroascorbate reductase (MDAR). These data are shown in Table 1. Following infection, the activity of POD, CAT and GST increased significantly in both genotypes, while the SOD activity increased significantly in DR genotype rather than DS genotype. APX and MDAR activity in DS genotype were decreased significantly by S. sclerotiorum infection, while DR genotype showed no significant change. S. sclerotiorum infection, caused higher activity of the enzymes relative to the DS genotype.”

Page 12 line 7

Replace “validate” with “validating”

Page 12 lines 18-24

The sentence in the beginning of the discussion section is poorly written and confusing. Please revise the sentence or consider removing it.

Reviewer #2: The presented study performed pairwise DEG analyses within susceptible and resistant genotypes, resulting in lists of genes and general pathways that may be involved in the sunflower defense response against the pathogen Sclerotinia sclerotiorum. Some main concerns are also listed below:

(1) There are new RNA-seq data generated, but the raw reads are unavailable.

(2) The findings are too general, with little information to conduct further studies or cross validations.

(3) There are expression profiles measured for susceptible and resistant genotypes, but functional/enrichment analyses are mainly focused on DEGs within genotypes. This will mainly lead to functions or pathways that are common between the two different genotypes. Related analyses for uniquely up/down-regulated genes in the resistant genotypes is recommended to identify candidate resistance genes that contribute to the resistance in the DR genotype compared with DS genotype. The analysis described in line 9-16 of page 10 is not comprehensive.

(4) The data processing parts in the method section needs to be revised to add detail information, including software and parameters used. For example, using "BMKCloud platform" does not provide sufficient information, and MapMan is a software instead of an analysis. A lot of information is missing.

(5) The genome assembly information is missing, which is critical for further use and validation of this study. Besides, the gene ID does not sound correct or published ones. For example, "Helianthus_annuus_newGene_2831" in table S2 and "NewGene_11665" in Figure 5B.

Other minor concerns:

(1) There should be a figure for mock-inoculated sample in Figure 1, which only presents inoculated leaves.

(2) Nearly a half of expressed genes are differentially expressed, which seems too many. The author may want to double check their data. For example, compare their result to previously published studies.

(3) The reads mapping ratio ranges from 74.8% to 91.8%. This imbalanced mapping ratio may contribute to the super-high number of DEGs.

6. PLOS authors have the option to publish the peer review history of their article (what does this mean?). If published, this will include your full peer review and any attached files.

Reviewer #1: No

Reviewer #2: No

---

## [Author Response · Author response to Decision Letter 0]

12 Sep 2023

Journal Requirements

1.Please ensure that your manuscript meets PLOS ONE's style requirements, including those for file naming. 

Answer: The manuscript was revised according to the PLOS ONE's style requirements. 

2.Thank you for stating the following financial disclosure: The study was supported by China Agriculture Research System of MOF and MARA (CARS-14), and Shenyang Seed Industry Innovation Project (22-318-2-19). Please state what role the funders took in the study.

Answer: We have stated that "The funders had no role in study design, data collection and analysis, decision to publish, or preparation of the manuscript." 

Answer: the GEO Series accession number GSE220161 (https://www.ncbi.nlm.nih.gov/geo/query/acc.cgi?acc=GSE220161) was added in the “Availability of data and materials”.

Answer: the corresponding author have an ORCID iD and that it is validated in Editorial Manager.

Answer: the Supporting Information files have been listed at the end of your manuscript.

Reviewer #1

Comments to the Author

The paper provides good insights into transcriptional changes occurring in susceptible and resistant sun flower following Sclerotinia sclerotiorum infection. There is a good attempt to measure physiological and correlate it with infection. The authors also provide a good model and graphical abstract as figure 8. For this, I commend the authors. Having said that, my biggest sticking point with the paper is the poor writing. It was difficult to understand some parts of the paper because of the way poor English and grammar. I pointed out a few and suggested revisions, but the whole paper will need to go through a thorough editing process. I was also less impressed by the quality of figures 4 and 5.

Answer: Accepted the suggestion, we have improved the language in this paper. In addition, the figures 4 and 5 were deleted, and these data was listed in S7-8 Tables and Tables 2-3.

Below are my specific comments:

Line 3

S. sclerotiorum

Page 4 line 2

Delete “thus far”

Page 4 line 18-19

Rephrase as follows:

“Both DR genotype C6 and DS genotype B728 are oilseed sunflower inbred lines which were bred by Liaoning Academy of Agricultural Sciences”

Line 23

Italicize “S. Sclerotiorum”

Line 5 Page 5

Replaced “were washing” with “washed”

Page 7 line 2 and throughout the paper,

Italicize “S. sclerotiorum”

Page 8 lines 3-12

Consider rephrasing as follows:

“The physiological response of leaves to S. sclerotiorum inoculation was further evaluated based on the activity of antioxidant enzymes, such as superoxide dismutase activity (SOD), peroxidase (POD), catalase (CAT), glutathione-S-transferase (GST), ascorbate peroxidase (APX) and monodehydroascorbate reductase (MDAR). These data are shown in Table 1. Following infection, the activity of POD, CAT and GST increased significantly in both genotypes, while the SOD activity increased significantly in DR genotype rather than DS genotype. APX and MDAR activity in DS genotype were decreased significantly by S. sclerotiorum infection, while DR genotype showed no significant change. S. sclerotiorum infection, caused higher activity of the enzymes relative to the DS genotype.”

Page 12 line 7

Replace “validate” with “validating”

Page 12 lines 18-24

The sentence in the beginning of the discussion section is poorly written and confusing. Please revise the sentence or consider removing it.

Answer: Accepted the suggestion, we have modified in revised manuscript.

Reviewer #2: 

The presented study performed pairwise DEG analyses within susceptible and resistant genotypes, resulting in lists of genes and general pathways that may be involved in the sunflower defense response against the pathogen Sclerotinia sclerotiorum. Some main concerns are also listed below:

(1) There are new RNA-seq data generated, but the raw reads are unavailable.

Answer: The raw reads were upload in NCBI, and the GEO Series accession number was GSE220161 (https://www.ncbi.nlm.nih.gov/geo/query/acc.cgi?acc=GSE220161)

(2) The findings are too general, with little information to conduct further studies or cross validations.

Answer: Accepted the suggestion. By RNA-seq analysis, we identified some genes involved in expansins, pectate lyase activities, ethylene biosynthesis and signaling and antioxidant activity, which showed differential expression between DS and DR genotypes. These could potentially be responsible for the differential resistance among two genotypes. However, it was difficulty to identify the candidate genes only via the RNA-seq analysis. 

 (3) There are expression profiles measured for susceptible and resistant genotypes, but functional/enrichment analyses are mainly focused on DEGs within genotypes. This will mainly lead to functions or pathways that are common between the two different genotypes. Related analyses for uniquely up/down-regulated genes in the resistant genotypes is recommended to identify candidate resistance genes that contribute to the resistance in the DR genotype compared with DS genotype. The analysis described in line 9-16 of page 10 is not comprehensive.

Answer: Accepted the suggestion. The GO analysis for the uniquely up- and down- regulated genes in DR genotype was conducted, and the results were listed in S7 and S8 tables. Also, the top twenty uniquely up- and down- regulated genes in DR genotype was listed in Table 2 and 3, and these functions were annotated. These results were analysized in the revised manuscript.

(4) The data processing parts in the method section needs to be revised to add detail information, including software and parameters used. For example, using "BMKCloud platform" does not provide sufficient information, and MapMan is a software instead of an analysis. A lot of information is missing.

Answer: Accepted the suggestion. We added the detail information in the method section for data processing parts.

(5) The genome assembly information is missing, which is critical for further use and validation of this study. Besides, the gene ID does not sound correct or published ones. For example, "Helianthus_annuus_newGene_2831" in table S2 and "NewGene_11665" in Figure 5B.

Answer: In the transcriptome analysis, it is normal to find new genes, which are not genes in the reference genome, and we will number and annotate the new genes.

Other minor concerns:

(1) There should be a figure for mock-inoculated sample in Figure 1, which only presents inoculated leaves.

Answer: Accepted the suggestion, the figures for mock-inoculated sample were shown in Figure 1.

(2) Nearly a half of expressed genes are differentially expressed, which seems too many. The author may want to double check their data. For example, compare their result to previously published studies.

Answer: We have check our data again. A total of 36969 and 37854 genes were expressed in mock-inoculated samples of DS and DR genotypes, respectively, while 36363 and 39538 in the S. sclerotiorum-inoculated samples.

(3) The reads mapping ratio ranges from 74.8% to 91.8%. This imbalanced mapping ratio may contribute to the super-high number of DEGs.

Answer: In this study, only the mapping ratio of sample of S. sclerotiorum-inoculated B728 was about 70%, which may be caused by the special treatment such as S. sclerotiorum infection.

---

## [Decision Letter · Decision Letter 1]

17 Mar 2024

PONE-D-23-14214R1Comparative Transcriptome Analysis in Two Contrasting Genotypes for Sclerotinia Sclerotiorum  Resistance in SunflowerPLOS ONE

Dear Dr. Liu,

Thank you for submitting your manuscript to PLOS ONE. After careful consideration, we feel that it has merit but does not fully meet PLOS ONE’s publication criteria as it currently stands. Therefore, we invite you to submit a revised version of the manuscript that addresses the points raised during the review process.

 I've read your current version of the manuscript and your responses to the reviewers. Together with the two external reviewers, this manuscript still needs major changes. The analyses need to be fully described for reproducibility (this was a main issue last round as well). The clarity of writing should be improved for readability.

We look forward to receiving your revised manuscript.

Kind regards,

Asela J. Wijeratne, Ph.D.

Academic Editor

PLOS ONE

Reviewers' comments:

Reviewer's Responses to Questions

**Comments to the Author**

1. If the authors have adequately addressed your comments raised in a previous round of review and you feel that this manuscript is now acceptable for publication, you may indicate that here to bypass the “Comments to the Author” section, enter your conflict of interest statement in the “Confidential to Editor” section, and submit your "Accept" recommendation.

Reviewer #2: (No Response)

Reviewer #3: (No Response)

2. Is the manuscript technically sound, and do the data support the conclusions?

Reviewer #2: Partly

Reviewer #3: Partly

3. Has the statistical analysis been performed appropriately and rigorously? 

Reviewer #2: No

Reviewer #3: No

4. Have the authors made all data underlying the findings in their manuscript fully available?

Reviewer #2: No

Reviewer #3: Yes

5. Is the manuscript presented in an intelligible fashion and written in standard English?

Reviewer #2: Yes

Reviewer #3: No

6. Review Comments to the Author

Reviewer #2: Please add the NCBI BioProject ID ‘PRJNA908908’ in the manuscript.

According to figure 4 B-E (figure 6 in previous version), nearly most genes are differentially expressed in both DS and DR genotypes (|log2(FC)| > 1), although the fold change varied. The expression of these genes was not statistically compared between DS and DR genotypes to narrow down the candidates, which could have been inferred from this data.

The read cleaning method is unclear and irreproducible. For example, a read should not be ‘annotated based on the reference genome’. This revised method section still contains many errors, making the results unreliable.

The reference genome information is still not provided, making it impossible to assess the results. Not only the ‘new genes’, but also the ‘reference genes’ have only an ‘ID’ available, which largely limits the contribution to the community.

The numbers of expressed genes were previously 22,077 and 22,579, but in the present version, they increased a lot. Again, without the information of the reference genome, it is impossible to know what happened.

Without sufficient data, I cannot comment on the numbers listed.

Reviewer #3: Authors have conducted transcriptome analysis of a resistant and a susceptible genotype of sunflower against Sclerotinia sclerotiorum. I have some suggestions and comments, which are given below. Page numbers are based on the revised, track-change version.

# Major comments:

1) Figure-4, S1, and S2 are not legible. Hence, I couldn’t check the information associated with these figures. These figures need to be updated.

2) One major criticism is that all the follow-up analyses were done with the DEGs (Differentially Expressed Genes) upon infection in both the resistant and the susceptible genotypes. This was pointed out by one of the reviewers. However, the author didn’t do a good job of resolving the issue. Ideally, it would be better to focus on DEGs that are unique to either the resistant or the susceptible genotype. Alternatively, performing clustering of DEGs could have been better.

3) English is very poor in some instances. For example, on page 17, lines 2-3, sentence construction is not correct. The same is true for line-8-9. There are multiple examples like this throughout the manuscript. Correcting those will improve the readability of the manuscript.

4) Did you use any filtering to get rid of the low-expressed genes, otherwise there will be DEGs that were expressed barely.

# Minor comments:

1) Usage of abbreviations should be kept at a minimum level. CWDE for ‘cell wall degrading enzyme’ (page 3, line-11) is unacceptable.

2) Page-5, line-6, why 24 hpi was used for the experiment?

3) Page 6, line 20, how the raw reads were processed?

4) What kind of analysis was performed using MapMan? MapMan is a software, not an analysis.

5) Page 8, line 27, what are the criteria for being expressed?

6) GO analysis of all DEGs is meaningless, authors should have focused on unique DEGs.

7) Page-9, line-23, “In the CC category, “cell”, “cell part”……were most frequent”. Is this kind of information meaningful?

8) Common GOs for both genotypes represented in Figure-3, are not very informative.

9) Page 10, lines 11-18, describe GO terms associated with the unique DEGs for each genotype. This section is useful. Authors should expand this section.

10) The result section for the analysis using MapMan is very rudimentary.

7. PLOS authors have the option to publish the peer review history of their article (what does this mean?). If published, this will include your full peer review and any attached files.

Reviewer #2: **Yes: **Honghe Sun

Reviewer #3: **Yes: **Manohar Chakrabarti

---

## [Author Response · Author response to Decision Letter 1]

3 May 2024

Reviewer #2:

Please add the NCBI BioProject ID ‘PRJNA908908’ in the manuscript.

Response: All the raw data of the transcriptome sequence have been deposited in the NCBI BioProject database (https://www.ncbi.nlm.nih.gov/bioproject/PRJNA908908) under the BioProject PRJNA908908.

According to figure 4 B-E (figure 6 in previous version), nearly most genes are differentially expressed in both DS and DR genotypes (|log2(FC)| > 1), although the fold change varied. The expression of these genes was not statistically compared between DS and DR genotypes to narrow down the candidates, which could have been inferred from this data.

Response: The expression of these genes was compared between DS and DR genotypes, which were shown in Fig. 4(A-D), and the results section.

The read cleaning method is unclear and irreproducible. For example, a read should not be ‘annotated based on the reference genome’. This revised method section still contains many errors, making the results unreliable.

Response: Raw reads of fastq format were firstly processed through in-house perl scripts. In this step, the clean reads were obtained by removing adapter-containing reads, ploy-N-containing reads, and low quality reads from raw data. The Q20, Q30, GC-content and sequence duplication level of the clean data were calculated. All the downstream analyses were based on clean data with high quality. The adaptor sequences and low-quality sequence reads were removed from the data sets. Raw sequences were transformed into clean reads after data processing. These clean reads were then mapped to the reference genome sequence. Only reads with a perfect match or one mismatch were further analyzed and annotated based on the sunflower reference genome. The HanXRQr1.0 genome was used as reference in the present study. Then, Hisat2 tools soft were used to map with reference genome. The StringTie Reference Annotation Based Transcript (RABT) assembly method was used to construct and identify both known and novel transcripts from Hisat2 alignment results.

The reference genome information is still not provided, making it impossible to assess the results. Not only the ‘new genes’, but also the ‘reference genes’ have only an ‘ID’ available, which largely limits the contribution to the community.

Response: The HanXRQr1.0 genome was used as reference in the present study.

The numbers of expressed genes were previously 22,077 and 22,579, but in the present version, they increased a lot. Again, without the information of the reference genome, it is impossible to know what happened.

Response: After careful inspection again，in the present version, the numbers of expressed genes were false, and those in the previous version were correct.

Without sufficient data, I cannot comment on the numbers listed.

Reviewer #3:

Authors have conducted transcriptome analysis of a resistant and a susceptible genotype of sunflower against Sclerotinia sclerotiorum. I have some suggestions and comments, which are given below. Page numbers are based on the revised, track-change version.

# Major comments:

1) Figure-4, S1, and S2 are not legible. Hence, I couldn’t check the information associated with these figures. These figures need to be updated.

Response: These figures were modified. 

2) One major criticism is that all the follow-up analyses were done with the DEGs (Differentially Expressed Genes) upon infection in both the resistant and the susceptible genotypes. This was pointed out by one of the reviewers. However, the author didn’t do a good job of resolving the issue. Ideally, it would be better to focus on DEGs that are unique to either the resistant or the susceptible genotype. Alternatively, performing clustering of DEGs could have been better.

Response: In GO analysis, the unique up- or down-regulated DEGs in either DS genotype B728 or DR genotype C6.

3) English is very poor in some instances. For example, on page 17, lines 2-3, sentence construction is not correct. The same is true for line-8-9. There are multiple examples like this throughout the manuscript. Correcting those will improve the readability of the manuscript.

Response: According the suggestion, these sentences were modified.

4) Did you use any filtering to get rid of the low-expressed genes, otherwise there will be DEGs that were expressed barely.

Response: Genes with low expression (FPKM <1) will be eliminated by default in the analysis process.

# Minor comments:

1) Usage of abbreviations should be kept at a minimum level. CWDE for ‘cell wall degrading enzyme’ (page 3, line-11) is unacceptable.

Response: Many abbreviations were deleted in the revised manuscript, such as for oxalic acid, cell-wall-degrading enzymes, programmed cell death, systemic acquired resistance, programmed cell death, pathogenesis related proteins, pathogen-associated molecular pattern.

2) Page-5, line-6, why 24 hpi was used for the experiment?

Response: As the visible lesions appeared in DS genotype rather than DR genotype at 24 h post-inoculation, then the 20 leaves samples within the range of 0.5 cm bordering the extending lesion from inoculated and mock-inoculated plants were harvested for each replicate.

3) Page 6, line 20, how the raw reads were processed?

Response: Raw reads of fastq format were firstly processed through in-house perl scripts. In this step, the clean reads were obtained by removing adapter-containing reads, ploy-N-containing reads, and low quality reads from raw data. The Q20, Q30, GC-content and sequence duplication level of the clean data were calculated. All the downstream analyses were based on clean data with high quality. 

4) What kind of analysis was performed using MapMan? MapMan is a software, not an analysis.

Response: According the suggestion, the MapMan analysis was modified as MapMan software.

5) Page 8, line 27, what are the criteria for being expressed?

Response: FPKM>1.

6) GO analysis of all DEGs is meaningless, authors should have focused on unique DEGs.

Response: The GO analysis of all DEGs was deleted in the revised manuscript.

7) Page-9, line-23, “In the CC category, “cell”, “cell part”……were most frequent”. Is this kind of information meaningful?

Response: The GO analysis of all DEGs was deleted in the revised manuscript.

8) Common GOs for both genotypes represented in Figure-3, are not very informative.

Response: The Figure-3 in previous version was deleted, and the S7-8 Tables were added in the present version. The S7-8 Tables were the GO analysis to identify enriched gene ontologies among the common DEGs in DS and DR genotypes.

9) Page 10, lines 11-18, describe GO terms associated with the unique DEGs for each genotype. This section is useful. Authors should expand this section.

Response: According the suggestion, the unique up- and down-regulated DEGs in DS genotype B728 were used for GO analysis, which were also shown in S9-10 Tables.

10) The result section for the analysis using MapMan is very rudimentary.

Response: According another reviewer’s suggestion, in the pathway analysis of DEGs using MapMan, the expressions of these genes involved in plant hormone signaling pathways, redox homeostasis and cell wall organisation were compared between DS and DR genotypes, which were shown in Fig. 4(A-D), and the results section.

---

## [Decision Letter · Decision Letter 2]

9 Sep 2024

PONE-D-23-14214R2Comparative Transcriptome Analysis in Two Contrasting Genotypes for Sclerotinia Sclerotiorum  Resistance in SunflowerPLOS ONE

Dear Dr. Liu,

Thank you for submitting your manuscript to PLOS ONE. After careful consideration, we feel that it has merit but does not fully meet PLOS ONE’s publication criteria as it currently stands. Therefore, we invite you to submit a revised version of the manuscript that addresses the points raised during the review process.

 I've read your current version of the manuscript and your responses to the reviewers. As the external reviewer noted, the manscript still require editing. Please go through reviewer's suggestion carefully.

We look forward to receiving your revised manuscript.

Kind regards,

Asela J. Wijeratne, Ph.D.

Academic Editor

PLOS ONE

Journal Requirements:

Additional Editor Comments:

I've read your current version of the manuscript and your responses to the reviewers. As the external reviewer noted the manscript still require editing. Please go through reviewers suggestion carefully.

Reviewers' comments:

Reviewer's Responses to Questions

**Comments to the Author**

1. If the authors have adequately addressed your comments raised in a previous round of review and you feel that this manuscript is now acceptable for publication, you may indicate that here to bypass the “Comments to the Author” section, enter your conflict of interest statement in the “Confidential to Editor” section, and submit your "Accept" recommendation.

Reviewer #1: (No Response)

2. Is the manuscript technically sound, and do the data support the conclusions?

Reviewer #1: (No Response)

3. Has the statistical analysis been performed appropriately and rigorously? 

Reviewer #1: (No Response)

4. Have the authors made all data underlying the findings in their manuscript fully available?

Reviewer #1: (No Response)

5. Is the manuscript presented in an intelligible fashion and written in standard English?

Reviewer #1: (No Response)

6. Review Comments to the Author

Reviewer #1: The ms has improved but still require editorial attention. Please see attached document with comments.

Figures are also low resolution - especially the supplementary figures that show enrichment. Labels on the X-axis have small fonts that are pixelated.

7. PLOS authors have the option to publish the peer review history of their article (what does this mean?). If published, this will include your full peer review and any attached files.

Reviewer #1: **Yes: **Steven Runo

---

## [Author Response · Author response to Decision Letter 2]

16 Sep 2024

Reviewer #1: The ms has improved but still require editorial attention. Please see attached document with comments.

Figures are also low resolution - especially the supplementary figures that show enrichment. Labels on the X-axis have small fonts that are pixelated.

Response: According another reviewer’s suggestion, the manuscript was improved.

---

## [Editor Report · Decision Letter 3]

26 Nov 2024

Comparative Transcriptome Analysis in Two Contrasting Genotypes for Sclerotinia Sclerotiorum  Resistance in Sunflower

PONE-D-23-14214R3

Dear Dr. Liu,

We’re pleased to inform you that your manuscript has been judged scientifically suitable for publication and will be formally accepted for publication once it meets all outstanding technical requirements.

Kind regards,

Asela J. Wijeratne, Ph.D.

Academic Editor

PLOS ONE

Additional Editor Comments (optional):

The authors have addressed all comments and issues raised by the reviewers, and the manuscript has been updated accordingly. However, there are still some editorial and grammatical mistakes that need to be corrected for the final submission. I kindly request you to carefully review the manuscript, identify any remaining issues, and make the necessary corrections.
---

## [Editor Report · Acceptance letter]

10 Dec 2024

PONE-D-23-14214R3 

PLOS ONE

Dear Dr. Liu, 

I'm pleased to inform you that your manuscript has been deemed suitable for publication in PLOS ONE. Congratulations! Your manuscript is now being handed over to our production team.

Kind regards, 

on behalf of

Dr. Asela J. Wijeratne 

Academic Editor

PLOS ONE